# Spatial Data-Based Automatic and Quantitative Approach in Analyzing Maintenance Reachability

**Jie Geng** [1], **Ying Li** [1,*], **Hailong Guo** [2], **Huan Zhang** [1] and **Chuan Lv** [1]

1  School of Reliability and Systems Engineering, Beijing University of Aeronautics and Astronautics, 37#, Xueyuan Road, Haidian District, Beijing 100191, China
2  AECC Hunan Aviation Powerplant Research Institute, Zhuzhou 412002, China
*  Correspondence: liying@buaa.edu.cn

**Abstract:** Reachability, as a vital parameter in product maintainability design, exerts a tremendous influence in practical maintenance, especially in the usage stage. To decrease subjectivity in maintenance reachability analysis, this study proposes an automatic and quantitative approach based on the spatial data of the human arm to implement maintenance reachability analysis. The approach focused on two aspects, namely, accuracy and efficiency. In terms of accuracy, the presented methodology starts from the maintenance spot where the human hand is attached. An original global data sequence set was generated, including the wrist, elbow, and shoulder joints, under the constraints of kinematics, in which a data sequence represents an arm motion. Moreover, the surrounding objects are represented by their geometric data, in which each data sequence is analyzed to judge whether collision occurs between arm segments and surrounding objects. In this filtering process, the data sequence is retained if the aforementioned collision does not occur. In terms of efficiency, owing to the large number of global data sequences, the efficiency of the interval selection in collision calculation is also taken into consideration in this methodology. Unlike the traditional methods in the virtual environment, the starting point is the maintenance spot, rather than the human body. Hence, nearly all possibilities of arm postures are considered in a global perspective with little subjective involvement, which enhances the automation and objectivity in maintenance reachability analysis to a certain extent. The case study shows the usability and feasibility by a practical maintenance scene.

**Keywords:** maintainability; maintenance reachability; quantitative analysis

## 1. Introduction

Owing to usage, transportation, and storage, failures ultimately happen even if products were at a high reliability level. The increasing complexity and powerful function of products may bring high requirements in the maintenance process. Hence, the well-designed maintainability is of particular importance for enhancing maintenance convenience and decreasing maintenance cost and duration.

Maintainability, as a vital element in product design, covers qualitative and quantitative factors. The qualitative section generally includes visibility, reachability, simplification, standardization, error proofing, ergonomics, and safety [1]. Among these elements, reachability is the major concern to users because "reaching" the target is always the first issue in maintenance steps.

Reachability represents how easily an object can be reached and maintained, in which a rational and logical reachability design makes a repair man approach the maintenance spot rapidly and conduct maintenance work conveniently [1,2]. The duration and cost can be effectively reduced further. A dissatisfactory reachability design not only increases maintenance difficulty, but also increases the cost and duration [3]. Consequently, evaluating reachability is extremely fundamental at the early stage of product design [4], and design flaw exposure is considerably ahead of its production under the support of the virtual prototype, whereas limitations, such as human position selection and human posture

setting, also bring subjective results [5]. The current reachability analysis methods can be roughly divided into three types: physical prototype-based, virtual prototype-based, and geometric calculation-based.

First, the physical prototype-based approach is conducted by operating real physical prototypes under maintenance rules and analyzing the extent of reachability. Although the analysis result approximates actual specifications, this strong dependency on a physical prototype always leads to hysteresis [6]. Hence, reachability design flaws need to be exposed after a product is produced and assembled. Aside from the high cost and time required, this hysteresis always results in incomplete improvement at the design stage, in which even flaws are exposed. Thus, inconvenient maintenance always occurs after a product is in use.

Second, the virtual prototype-based approach reveals a product in advance for users, and the corresponding reachability analysis can be conducted in a virtual environment on the basis of virtual reality technology and necessary analysis tools [7]. In this approach, design flaws are exposed and improved in a predictable manner. The procedure of reachability analysis based on virtual prototypes is summarized as follows. Digital human modeling technology [8–10], an essentially basic technology in virtual maintenance, is firstly introduced to generate different specifications of human body models [11,12], and based on the digital model, a shoulder joint-based envelope that approximates a sphere is established to present the reachability scope [13]. This envelope moves along with the human position or posture changes. In the virtual environment, once the position or posture of the virtual human is confirmed, the corresponding envelope is generated to show the current reachability scope, and the objects in this envelope are treated as reachable. On the contrary, objects outside the envelope are regarded as unreachable. This envelope-based approach is widely used in virtual maintenance to verify reachability, owing to its intuitive visualization by means of this envelope. Although taken seriously, envelope establishment relies on a fixed posture, while the fixed posture also depends on the subjectivity of the analyst, in that even though the analyst has considered a number of scenarios, the generated envelopes remain limited. Thus, the reachability analysis results are also limited. Although this approach provides a predictable and visual method, disregarding ignored positions and postures may also lead to an incomplete reachability analysis.

Third, the geometric calculation-based approach can be summarized in the following steps. The hand position and solution space of the shoulder joints' position are determined, then, several random shoulder joint positions are selected in this space. Next, the arm pose is calculated on the basis of constraints, such as joint angle, collision, and hand and shoulder joint positions. The maintenance spot is reachable if a set of solutions can be obtained, otherwise it is unreachable. This approach has a low dependence on the person, but it does not consider all arm poses because it can only obtain and evaluate limited solutions. It is also a local area-oriented analysis method.

From the view of kinematics, human arms, especially the hand, are the most flexible segments, owing to a series of joints, including the humerus, ulna, radius, carpi, metacarpi, phalanges, and corresponding muscles. Hence, the arms and hands, with a high participation, played a vital role in maintenance, wherein almost all maintenance activities rely on arms and hands. Consequently, the focus on arms and hands is functional and effective for the maintenance reachability analysis.

On the basis of the analysis above, avoiding the poor timeliness of the physical prototype-based way is the first issue to be resolved in obtaining a predictable and objective result in the maintenance reachability analysis. Decreasing the subjectivity of the virtual prototype-based method is another issue that needs to be addressed. In addition, accuracy and efficiency should also be concerns.

Consequently, by adopting the virtual prototype-based approach for a predictable analysis, as shown in Figure 1, this study proposes an objective and quantitative methodology to implement the maintenance reachability analysis. First, unlike the traditional methods that started from the human body, the presented methodology starts from the

maintenance spot where the human hand is attached. Second, to enhance objectivity, an original global data sequence set, including the wrist, elbow, and shoulder joints under the constraints of kinematics, is generated, in which a data sequence represents an arm motion by processing the spatial data based on the kinematic relationships among the joints of the human arm, using the position and rotation data of each joint. Third, in terms of accuracy, the surrounding objects are also represented by their geometric data. Each data sequence is analyzed to judge whether collision occurs between the arm segments and surrounding objects. In this filtering process, the data sequence is retained if the aforementioned collision does not occur. In terms of efficiency, owing to the large amount of global data sequences, the efficiency of the interval selection in the collision calculation is also taken into consideration in this methodology.

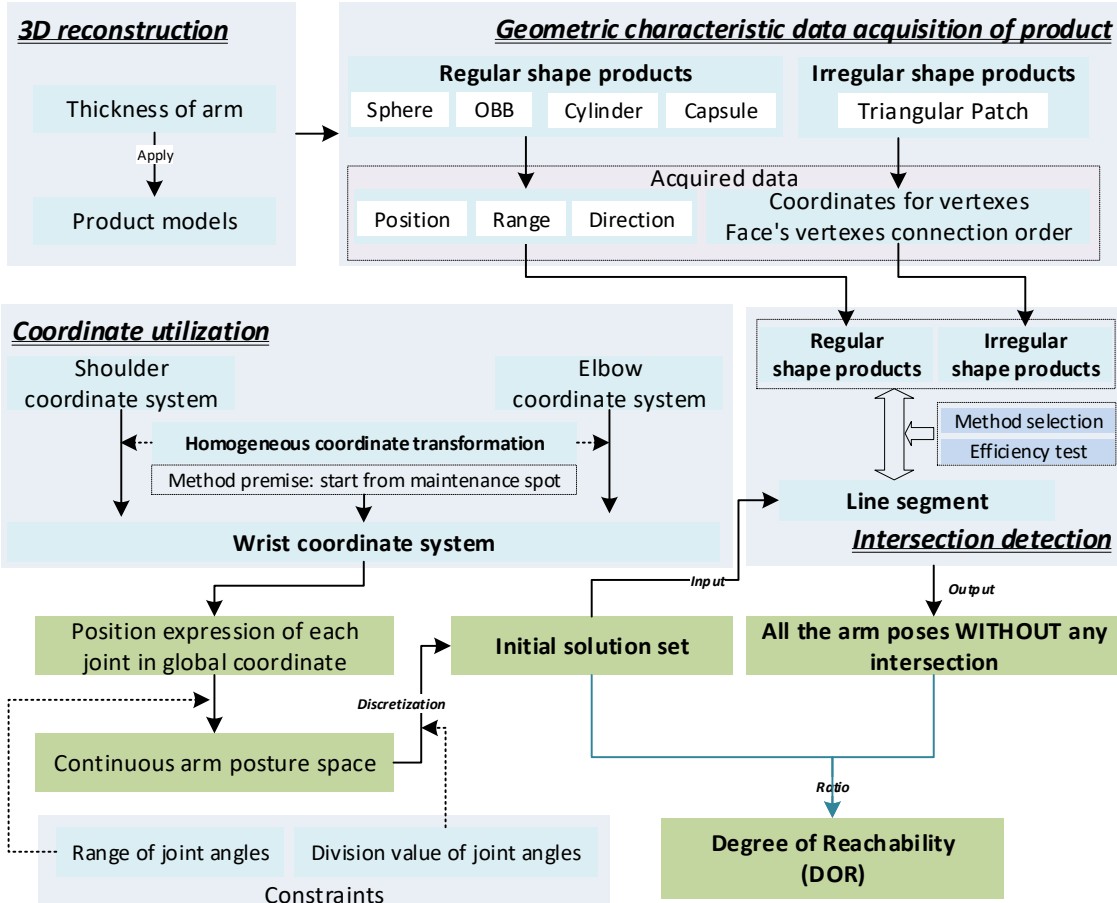

**Figure 1.** Overall Structure.

## 2. Literature Review

Recent publications that are relevant to the proposed method are concerned with three research streams: maintenance and maintainability, virtual maintenance technology, and maintenance reachability analysis. In this section, relevant studies were summarized.

For maintenance and maintainability, these two closely related concepts influence each other in practical engineering. Maintenance is a kind of work undertaken by maintenance personnel to maintain or restore the product to serviceable condition, and maintainability is a design character taken by designers in the development process to combine the features of improving the convenience of maintenance into the product design [14]. In order to facilitate maintenance, the maintainability design should start from the early stage and emphasize early intervention. Generally, maintainability requirements cover qualitative and quantitative sections. Specifically, qualitative design and analysis are conducted by maintainability check, and quantitative design and analysis are conducted by allocation

and prediction. In recent years, advanced maintenance technology has been introduced, for instance, the intelligent predictive maintenance is more in line with the industrial development trend, such as condition-based maintenance, prognostics and health management, and remaining useful life estimation [15], and this maintenance policy has been adopted by many industries in various domains. In this advanced maintenance method, intelligent sensors are extremely essential for the realization of predictive maintenance [16–18].

For virtual maintenance technology, a variety of studies have been conducted since the late 1990s. The advent of virtual reality technology provides better support for maintainability. The current research on virtual maintenance technology focuses on the following aspects: virtual maintenance prototype modeling [19–23], virtual simulation action process generation [24–26], maintenance training [27,28], product design [29,30], and maintainability evaluation under virtual simulation [31–33].

For virtual maintenance prototype modeling, an approach to the network-centric virtual prototyping in a distributed design environment was presented by Lee [21]. The approach has combined the virtual assembly modeling and analysis technique with distributed computing and communication technology for supporting the virtual prototyping activities over a network. A virtual maintenance system was developed and the flowchart of the model transformation technology was designed by Liu et al. [22]. Moreover, a virtual reality environment prototype of a building management system for maintenance activities was implemented by Carreira et al. [19].

For example, a low-cost VR application for maintenance training, which is an object-oriented prototype desktop VR-enabled system for maintenance (V-REALISM), is presented by Li et al. [28]. A low-complexity method is proposed to author an interactive virtual maintenance training system of hydroelectricity-generating equipment [27].

For instance, on the maintainability evaluation under virtual simulation, Lockheed Martin eliminated the metal models that had been used for design research and used virtual maintenance technology based on the CAD model to conduct an analysis on the maintainability evaluation, verification, and human factor when designing the fighter system of the F-16 [33]. A systematic approach for human factor automatic evaluation for the entire maintenance processes in the virtual environment was proposed by Qiu et al. [32]. A time prediction method of the assembly and disassembly of the product maintenance process was proposed by Cai et al. on the basis of virtual maintenance [31].

Recently, augmented reality technology has been widely used in maintainability evaluation and verification. However, the rock crusher repair platform was verified by the augmented maintenance technology and virtual maintenance technology by Aromaa and Väänänen [34]. As a result, virtual maintenance technology is more suitable for the evaluation and verification of visibility, reachability, and operability than augmented maintenance technology.

In the maintenance reachability analysis, the inverse kinematics and dynamics analysis of the human body is required. However, the human body is a complex, non-linear, and multiple redundancy system, which is difficult to be modeled, solved, and simulated. Therefore, this issue has been a research hotspot. Feasible technical approaches have emerged in recent years, and one of the most widely used is the envelope-based approach, owing to the kinematic characteristics of the human arm.

In terms of envelope establishment, various approaches for human reach envelopes were introduced by Yang and Abdel-Malek [35], such as the experiment-based approach, voxel-based method, and closed-form method. Yang, Sinokrot, and Abdel-Malek presented a general analytical method to determine the upper extremity workspace for any percentile virtual humans [36]. The upper extremity workspace and comfort index of the human body was obtained according to the changes in joint limits and limb length and the kinematics model. Chen et al. analyzed the influencing factors of maintenance reachability [37], whereas a mathematical model for the continuous quantification of the maintenance reachability based on the kinematic model of a human arm was established. Only the arm in the

free state was analyzed in this study, that is, the interference of objects in the surrounding environment was ignored.

In terms of the envelope application, Lee, Tsai, and Kang examined the problem of pipeline maintenance in architecture [37], and a visual tool for pipeline maintenance design was proposed to evaluate the visibility, reachability, and operability of the pipeline. In the reachability analysis section, a cylinder is used to envelop the person, and the cylinder movement is used to represent the path of the person. Moreover, an immersive virtual maintenance simulation system built by the Cave Automated Virtual Environment was introduced [38], which is a motion capture system and virtual simulation software. Using this system, the visibility, reachability, and other factors in the maintainability of products can be evaluated. In this method, maintenance reachability analysis is realized with the help of a shoulder joint-based envelope. Liu and Issa investigated the design for the maintenance accessibility method [4], whereas the maintenance accessibility design was accomplished using a tool in building information modeling software, which was called the Solibri Model Checker (SMC). The ruleset used to check the reachability in SMC is "free area in front of component", whose essence is also a reach envelope.

Moreover, optimization algorithms can be used in maintenance reachability analysis. Grignon and Fadel established a mathematical model for the layout optimization of the components in a small satellite cabin with maintenance reachability as one of the goals [39]. A multi-objective genetic algorithm was examined to solve this problem.

According to the investigation, the physical prototype-based method is poor in timeliness because it can only be conducted after a design is finished, and the virtual prototype-based method is subjective because it requires the analyst to operate the virtual person during the whole analysis process. The calculation-based method has the limitation of the solution because only a limited set of solutions can be obtained for evaluation.

Addressing the subjectivity and limitation of current maintenance reachability analyses based on virtual maintenance technology, the proposed methodology starts from the initial global solution set of the arm. By considering the constraints of arm joints and surrounding obstacles, the initial solution set is generated, optimized, and filtered sequentially to realize the quantitative analysis and representation of maintenance reachability.

### 3. Materials and Methods

*3.1. Generation of the Arm's Maximum Solution Set*

3.1.1. Arm Model Establishment

From the viewpoint of kinematics, arm movements are enabled by joints that drive the bones and this process can be regarded as the motion of spatial linkage. Hence, the model of the arm can be simplified as a spatial linkage mechanism. In the model, joints are equivalent to kinematic pairs, whereas bones are equivalent to links.

The methodology presented in this study evaluates reachability on the basis of hand fixation on the maintenance spot. Generally, once a hand is placed on the repair site or has grabbed the maintenance tool, the fingers' degrees of freedom (DOF) will barely affect reachability. In the proposed methodology, the fingers' DOF can be ignored and the hand can be simplified as a point. The established arm model includes seven DOFs, two of which are for the wrists, one is for the elbow, three are for the shoulders, and one is for the forearms. In this model, each DOF is equivalent to one link, in which the lengths of the forearm and upper arm are their actual values, whereas the lengths of other links are zero.

3.1.2. Position Expression of Each Joint in Global Coordinate

For the purpose of representing all joints in one coordinate system, a coordinate system for each link is established, and two adjacent coordinate systems are connected by a homogeneous transformation matrix. Therefore, the coordinates of one of the coordinate systems are transformed into the other. Consequently, the links in the model can be connected by corresponding homogeneous transformation matrices. The joints can be represented in the same coordinate system.

The simple mechanism shown in Figure 2 represents the attitude relationship between any near two joints.

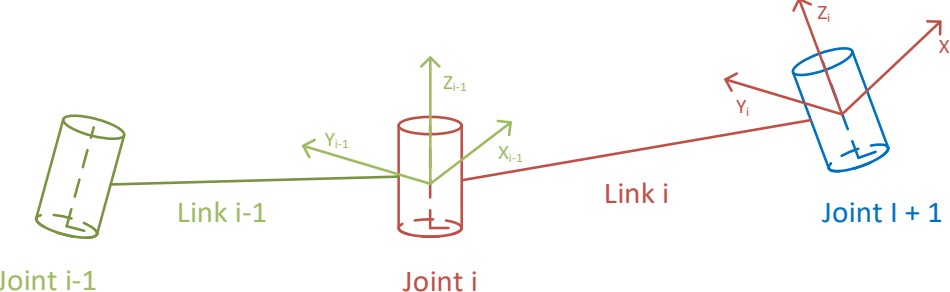

**Figure 2.** Relationship Between Two Adjacent Links.

The homogeneous transformation matrix, which was also called the D-H transformation matrix, between these two adjacent links is generally represented as $^{i-1}T_i$.

$$^{i-1}T_i = \begin{bmatrix} R_{3*3} & P_{3*1} \\ O_{1*3} & I_{1*1} \end{bmatrix} = \begin{bmatrix} \cos\theta_i & -\sin\theta_i\cos\alpha_i & \sin\theta_i\sin\alpha_i & a_i\cos\theta_i \\ \sin\theta_i & \cos\theta_i\cos\alpha_i & -\cos\theta_i\sin\alpha_i & a_i\sin\theta_i \\ 0 & \sin\alpha_i & \cos\alpha_i & d_i \\ 0 & 0 & 0 & 1 \end{bmatrix} \tag{1}$$

The meaning of each parameter in Equation (1) is:

(1)   $a_i$ represents the vertical distance between the axes of near two joints (z-axis) and the length of link $i$;
(2)   $\alpha_i$ represents the angle between the axes of near two joints (z-axis) and the twist angle of link $i$;
(3)   $d_i$ represents the distance between the vertical lines (x-axis) of near two joints' axes and the offset distance of link $i$ relative to link $i-1$;
(4)   $\theta_i$ represents the angle between the vertical lines (x-axis) of near two joints' axes and the rotation angle of link $i$ relative to link $i-1$.

This formula represents the transformation from the $i$ coordinate system to the $i-1$ coordinate system. In this formula:

$R_{3*3}$, $P_{3*1}$, $O_{1*3}$, $I_{1*1}$ represents the rotation matrix, position matrix, perspective matrix, and scale matrix, respectively;

$a_i$, $\alpha_i$ represents the vertical distance and the angle between the axes of near two joints (z-axis);

$d_i$, $\theta_i$ represents the distance and the angle between the vertical lines (x-axis) of near two joints' axes.

In the four matrixes above, the position matrix is the major concern in the proposed methodology.

The nth coordinate system can be transformed into the reference coordinate system by several D-H transformation matrixes:

$$^{0}T_n = {}^{0}T_1\,{}^{1}T_2\cdots{}^{n-1}T_n \tag{2}$$

According to the transition process, the coordinate systems in the arm model are unified into the reference coordinate system. Figure 3 shows the transition result by taking the hand as the reference coordinate system. Table 1 shows the required parameter values in the D-H transformation matrixes.

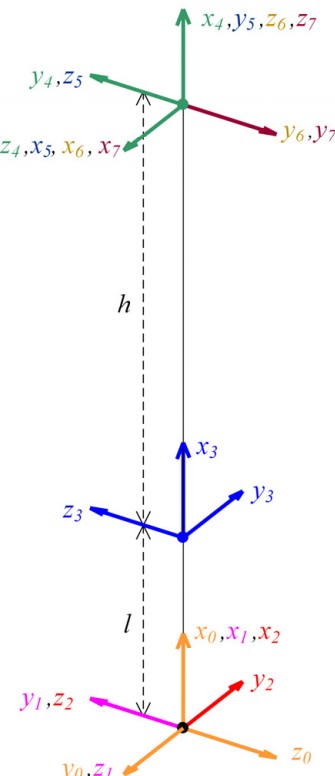

**Figure 3.** Arm Coordinate System.

**Table 1.** Arm's Parameters in D-H Transformation Matrices.

| $i$ | Joint Angles | Zero Position | $\alpha_i$ | $a_i$ | $d_i$ |
|---|---|---|---|---|---|
| 1 | $\theta_1$ | $-90°$ | $-90°$ | 0 | 0 |
| 2 | $\theta_2$ | $-90°$ | $-90°$ | 0 | 0 |
| 3 | $\theta_3$ | $-90°$ | $0°$ | $l$ | 0 |
| 4 | $\theta_4$ | $0°$ | $90°$ | $h$ | 0 |
| 5 | $\theta_5$ | $0°$ | $-90°$ | 0 | 0 |
| 6 | $\theta_6$ | $90°$ | $-90°$ | 0 | 0 |
| 7 | $\theta_7$ | $0°$ | $0°$ | 0 | 0 |

In Table 1, $i$ ranges from 1 to 7, which are the first DOF of the wrist joint; the DOF of the forearm; the second DOF of the wrist joint; the DOF of the elbow joint; and the first, second, and third DOF of the shoulder joint. $l$ is the length of the forearm, the value of which is 267 mm, and $h$ is the length of the upper arm, the value of which is 339 mm. $\theta_1$, $\theta_3$ are the rotational angles of the wrist joint; $\theta_2$ is the rotational angle of the forearm; $\theta_4$ is the rotational angle of the elbow joint; $\theta_5$, $\theta_6$, $\theta_7$ are the rotational angles of the shoulder joint. The ranges of $\theta_1$ to $\theta_7$ are $-135° \leq \theta_1 \leq -45°$, $-90° \leq \theta_2 \leq 70°$, $-175° \leq \theta_3 \leq 10°$, $0° \leq \theta_4 \leq 142°$, $0° \leq \theta_5 \leq 180°$, $45° \leq \theta_6 \leq 225°$, $-135° \leq \theta_7 \leq 90°$.

The variable is $\theta_i$, for convenience, this study uses $si$ for $\sin \theta_i$ and $ci$ for $\cos \theta_i$.

According to Formula (1), the D-H transformation matrices between each two coordinate systems can be obtained as:

$$
{}^{0}T_{1} = \begin{bmatrix} c1 & 0 & -s1 & 0 \\ s1 & 0 & c1 & 0 \\ 0 & -1 & 0 & 0 \\ 0 & 0 & 0 & 1 \end{bmatrix}, \quad {}^{1}T_{2} = \begin{bmatrix} c2 & 0 & -s2 & 0 \\ s2 & 0 & c2 & 0 \\ 0 & -1 & 0 & 0 \\ 0 & 0 & 0 & 1 \end{bmatrix}, \quad {}^{2}T_{3} = \begin{bmatrix} c3 & -s3 & 0 & lc3 \\ s3 & c3 & 0 & ls3 \\ 0 & 0 & 1 & 0 \\ 0 & 0 & 0 & 1 \end{bmatrix}
$$

$$
{}^{2}T_{3} = \begin{bmatrix} c3 & -s3 & 0 & lc3 \\ s3 & c3 & 0 & ls3 \\ 0 & 0 & 1 & 0 \\ 0 & 0 & 0 & 1 \end{bmatrix}, \quad {}^{5}T_{6} = \begin{bmatrix} c6 & 0 & -s6 & 0 \\ s6 & 0 & c6 & 0 \\ 0 & -1 & 0 & 0 \\ 0 & 0 & 0 & 1 \end{bmatrix}, \quad {}^{4}T_{5} = \begin{bmatrix} c5 & 0 & -s5 & 0 \\ s5 & 0 & c5 & 0 \\ 0 & -1 & 0 & 0 \\ 0 & 0 & 0 & 1 \end{bmatrix}
$$

$$
{}^{6}T_{7} = \begin{bmatrix} c7 & -s7 & 0 & 0 \\ s7 & c7 & 0 & 0 \\ 0 & 0 & 1 & 0 \\ 0 & 0 & 0 & 1 \end{bmatrix}
$$

The transformation matrix from the shoulder joint coordinate system to the base coordinate system ${}^{0}T_{7}$ and the transformation matrix from the elbow joint coordinate system to base coordinate system ${}^{0}T_{2}$ can be obtained as follows:

$$
{}^{0}T_{7} = {}^{0}T_{1}{}^{1}T_{2}{}^{2}T_{3}{}^{3}T_{4}{}^{4}T_{5}{}^{5}T_{6}{}^{6}T_{7} = \begin{bmatrix} R^{07}_{3*3} & P^{07}_{3*1} \\ O_{1*3} & I_{1*1} \end{bmatrix} \tag{3}
$$

$$
{}^{0}T_{3} = {}^{0}T_{1}{}^{1}T_{2}{}^{2}T_{3} = \begin{bmatrix} R^{03}_{3*3} & P^{03}_{3*1} \\ O_{1*3} & I_{1*1} \end{bmatrix} \tag{4}
$$

The formula $P^{07}_{3*1} = \begin{bmatrix} l*(s1*s3 + c1*c2*c3) + h*[s1*s(3+4) + c1*c2*c3*(c4-s4)] \\ l*(s1*c2*c3 - c1*s3) - h*[c1*s(3+4) - s1*c2*c(3+4)] \\ -l*s2*c3 - h*s2*c(3+4) \end{bmatrix}$ is the expression for shoulder joint coordinates;

The formula $P^{03}_{3*1} = \begin{bmatrix} l*s1*s3 + l*c1*c2*c3 \\ l*s1*c2*c3 - l*c1*s3 \\ -l*s2*c3 \end{bmatrix}$ is the expression for elbow joint coordinates.

Note: The expressions for shoulder joint coordinates and elbow joint coordinates are obtained in the case that the wrist coordinate is $(0, 0, 0)$.

According to the expressions for shoulder joint coordinates and elbow joint coordinates, different shoulder joint and elbow joint positions can be obtained by setting different values of $\theta_1$ to $\theta_7$. The forearm and upper arm can be represented by connecting the elbow joint to the corresponding wrist joint and the corresponding shoulder joint. The shoulder joint and elbow joint positions can represent different arm postures.

3.1.3. Discretization of the Continuous Arm Posture Space

According to the analysis, $\theta_5$, $\theta_6$, $\theta_7$ do not appear in the formula. The changes of $\theta_5$, $\theta_6$, $\theta_7$ will not affect the positions of the shoulder joint and elbow joint once the hand is fixed. Only $\theta_1$, $\theta_2$, $\theta_3$, $\theta_4$ can affect the positions of the shoulder joint and elbow joint.

Four sets can be obtained by taking the values of these four angles from the upper limit to the lower limit every n-degrees. These four sets are the values of $\theta_1$, $\theta_2$, $\theta_3$, $\theta_4$, which include $90/n$, $160/n$, $186/n$, $142/n$ angle values, respectively.

Taking the values from the four sets can obtain a set containing four angle values. Iterating through all the value combinations can obtain $N = \frac{90 \times 160 \times 186 \times 142}{n^4}$ different value sets. By placing these value sets into the expressions for shoulder joint coordinates and elbow joint coordinates, the elbow joint and shoulder joint positions that correspond to the value sets can be obtained. The arm pose can be expressed by the elbow joint and shoulder joint positions. In this way, the continuous arm posture change space can be discretized. These discrete solutions constitute the initial solution set of the arm pose that can move freely in the barrier-free space when the hand is fixed.

### 3.2. Maintenance Reachability Analysis-Oriented 3D Reconstruction of Maintenance Scene

3.2.1. Product Model Reconstruction

For the purpose of simplifying the judgment of the spatial relationship in the subsequent steps, the product models need to be reconstructed first. In the proposed methodology, the arm that was originally regarded as a cylinder is treated as a line segment, then the thickness of arm, which is the radius of cylinder, is superposed on the product models, as the surface of each product model expands outward by the same thickness, and the product models are reconstructed. In this way, the complex body-to-body intersection detection is translated into simpler line-to-body intersection detection. We used a spherical object as an example.

Figure 4 is a schematic diagram of judging the spatial relationship between a forearm and a spherical object. In this figure, the cylinder with radius $r_1$ represents the forearm and the sphere object with radius $R_2$ represents the product that needs to judge the spatial relationship with the forearm. The thickness of forearm $r_1$ is added to the sphere, as the surface of the sphere expands outward, $r_1$ and the sphere's radius becomes $r_1 + R_2$. The forearm is simplified from a cylinder to a line segment. The spatial relationship between the forearm and the sphere can be obtained by detecting the intersection between the expanded sphere and the line segment. Therefore, the detection efficiency will be improved by this reconstruction.

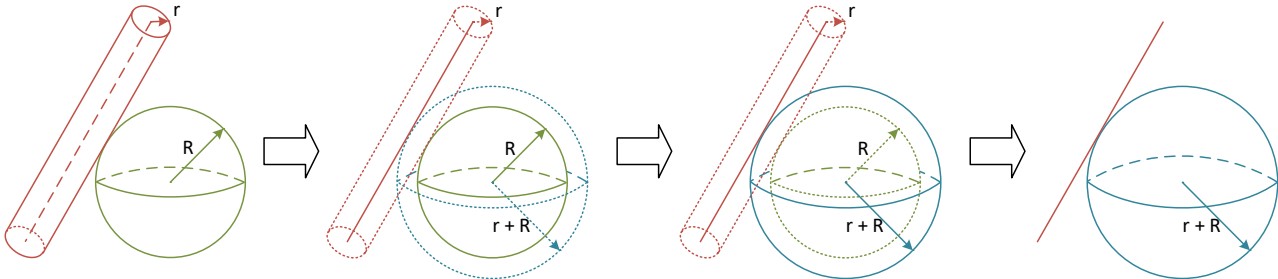

**Figure 4.** Arm Model Simplification Diagram.

During the judgements of the relationship between the arm and products, the forearm and the upper arm need to be considered simultaneously. Owing to the different thicknesses of these two segments, two corresponding calculation scenes should be built as follows. Assuming the thicknesses of the forearm and the upper arm are $r_1$ and $r_2$, respectively, each product's surface expands outwards, for $r_1$ and $r_2$ to conduct corresponding intersection detection with the forearm and the upper arm separately.

3.2.2. Geometric Characteristic Data Acquisition of Product

If the reconstructed product models are regular shapes, then they will be represented by bounding spheres, bounding boxes, cylinders, and capsules. Otherwise, they will be meshed and represented by triangular patches. The necessary geometric characteristic data of each product model should be obtained for subsequent calculation, as shown in Table 2.

**Table 2.** Necessary Geometric Characteristic Data of Different Product Models.

| Product Model | Geometric Characteristic Data | | |
| --- | --- | --- | --- |
| | **Position Class** | **Range Class** | **Direction Class** |
| Sphere | Center Coordinate | Radius | — |
| OBB | Center Coordinate | Range Vector (Half Length of Three Sides) | Local Axes (Direction Vector of Three Sides) |
| Cylinder | Axis Ends' Coordinates | Radius | — |
| Capsule | Axis Ends' Coordinates | Radius | — |
| Triangular Patch | Coordinates for Vertexes | — | Face's Vertexes Connection Order |

*3.3. Intersection Detection-Based Spatial Relationship Judgment between Arm and Product*

3.3.1. Application of Typical Intersection Detection Methods Illustration

According to the simplified models, one side of the intersection detection is the line segment and the other side covers five categories, namely, the bounding sphere, bounding box, cylinder, capsule, and triangle patch. Detailed descriptions of the five intersection detection processes are provided next.

(1)   Intersection detection of a line segment and a sphere

The intersection detection of a line segment and a sphere is relatively simple, that is, comparing distance $d$ from the center of the sphere to the line segment and the radius $R$ of the sphere. As shown in Figure 5, if $d > R$, then the line segment AB does not intersect with the sphere, and if $d < R$, then the line segment AB intersects with the sphere.

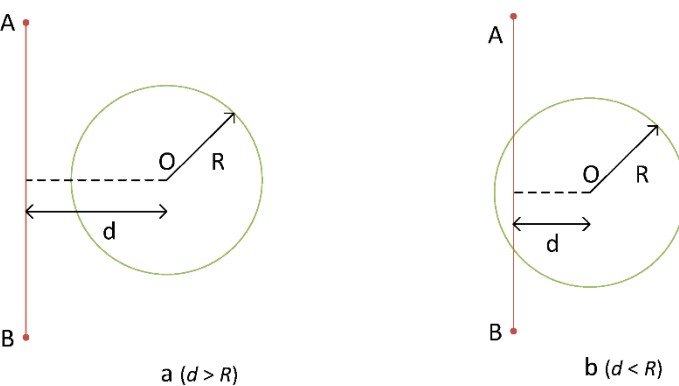

**Figure 5.** Diagram of the Intersection Detection of a Line Segment and a Sphere.

(2)   Intersection of a line segment and an oriented bounding box (OBB)

The intersection of a line segment and an OBB is detected by the separating axis theory. If a hyperplane $P$ exists, such that the line segment and the OBB are on both sides of the hyperplane, then the line segment does not intersect with the OBB, otherwise it intersects.

As shown in Figure 6, $r_b$ is the projection radius of the OBB on vector $v$, $r_s$ is the projection radius of line segment AB on vector $v$, and $d$ is the projected length of the distance between the center point of the OBB and the midpoint of line segment AB on vector $v$.

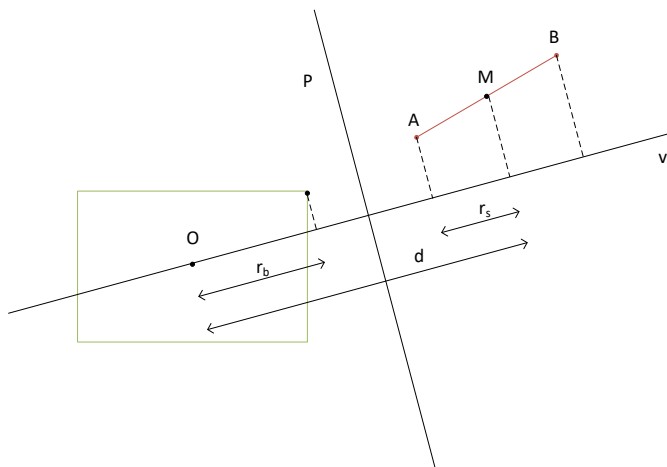

**Figure 6.** Diagram of the Intersection Detection of a Line Segment and an OBB.

If $r_b + r_s < d$, then vector $v$ is a separating axis of line segment AB and the OBB, and plane $P$ is a hyperplane, line-segment AB and the OBB are on both sides of plane $P$;

If $r_b + r_s > d$, then vector $v$ is not a separating axis of line segment AB and the OBB, and plane $P$ is not a hyperplane, line segment AB and the OBB are not on both sides of plane $P$;

The detection between the line segment and the OBB needs to be tested for six axes, which are three face normal of the OBB and three cross-product vectors of the three face normal and direction vector of line segment AB. As long as one axis is the separate axis, line segment AB does not intersect with the OBB, otherwise line segment AB intersects with the OBB.

(3)    Intersection of a line segment and a cylinder

Aside from the two intersection detection processes described above, the detection between the line segment and the cylinder is complex. The intersection between the line segment and the cylinder is detected by the method of finding the intersection point. If line segment AB does not have an intersection with the cylinder, then the line segment does not intersect with the cylinder, otherwise it intersects. Figure 7 shows the judgment process.

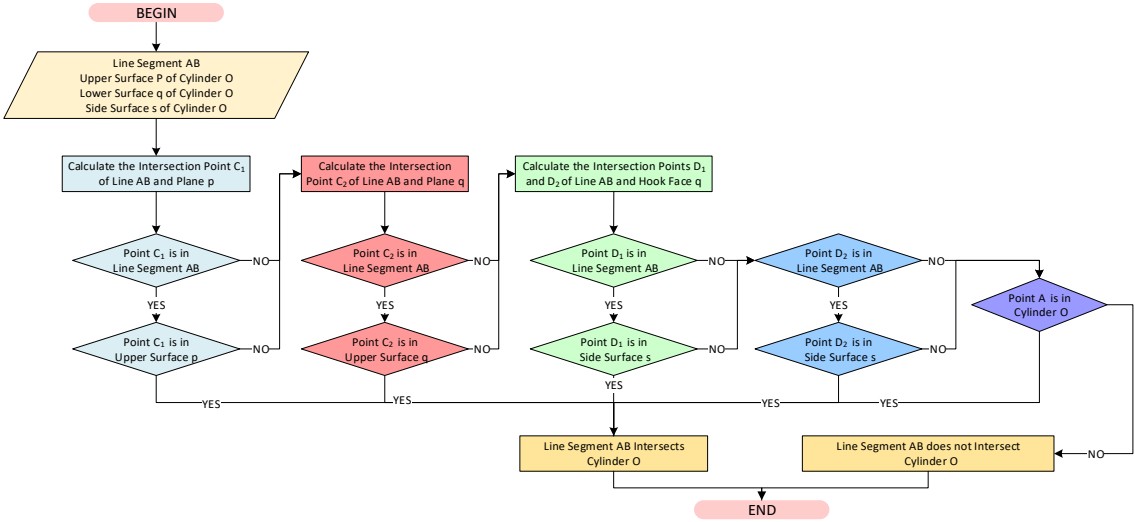

**Figure 7.** Application of the Intersection Detection Between a Line Segment and a Cylinder.

First, whether the line where line segment AB is located intersects with the plane where the upper surface of the cylinder is located should be determined. If it intersects and the intersection point is in the line segment and the upper surface of the cylinder, then the line segment intersects with the cylinder.

Otherwise, whether the line where line segment AB is located intersects with the surface where the lower surface of the cylinder is located should be determined. If it intersects and the intersection point is in the line segment and the lower surface of the cylinder, then the line segment intersects with the cylinder.

Otherwise, whether the line where line segment AB is located intersects with the surface where the side surface of the cylinder is located should be determined. If it intersects and the intersection point is in the line segment and the side surface of the cylinder, then the line segment intersects with the cylinder.

Otherwise, whether all line segments are inside the cylinder should be determined by selecting an end point for verification. If the end point is in the cylinder, then the line segment intersects with the cylinder. If all conditions are not met, then the line segment does not intersect with the cylinder.

As shown in Figure 8a–d, line segment AB intersects the bottom surface of the cylinder, line segment AB intersects with the side surface of the cylinder, line segment AB is inside the cylinder, and line segment AB does not intersect with the cylinder, respectively.

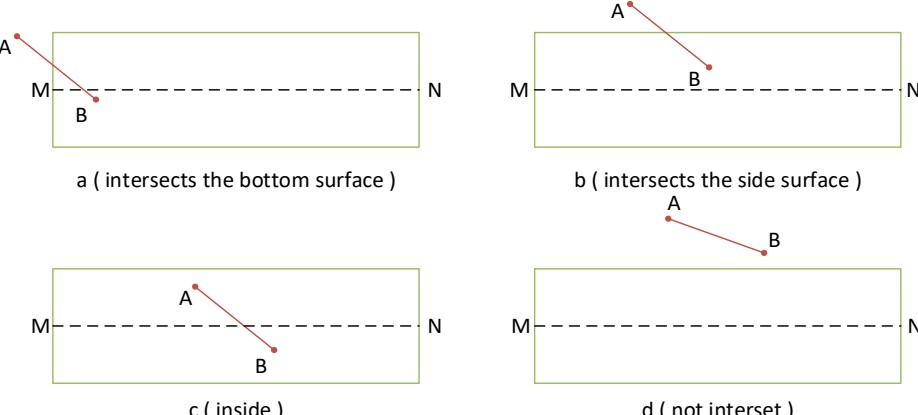

Figure 8. Diagram of the Intersection Detection of a Line Segment and a Cylinder.

(4) Intersection of a line segment and a capsule

The intersection of a line segment and a capsule is detected by comparing distance $d$ from the capsule axis to the line segment and the radius $R$ of the capsule. As shown in Figure 9, if $d > R$, then line segment AB does not intersect with the capsule, but if $d < R$, then line segment AB intersects with the capsule.

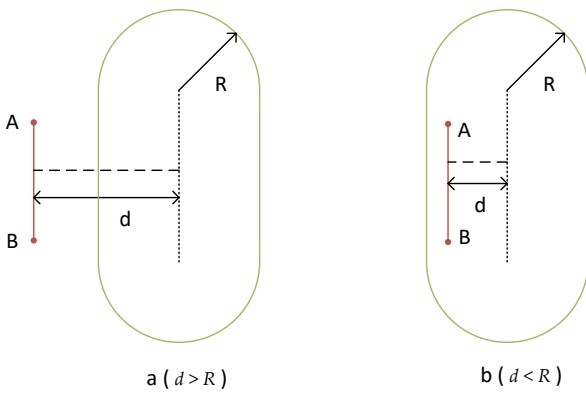

Figure 9. Diagram of the Intersection Detection of a Line Segment and a Capsule.

(5) Intersection of a line segment and a triangular patch

The intersection of a line segment and a triangular patch is detected by the method of finding the intersection point. Whether the line where the line segment is located would intersect with the plane where the triangular patch is located should be determined. If it intersects, then the intersection point would be located. Then, if the intersection point is in the line segment and the triangular patch, then the line segment intersects with the triangular patch, otherwise it does not intersect.

3.3.2. Solving the Maintenance Reachability Feasible Region

The maintenance reachability feasible region is represented by a solution set, which can be solved from the initial solution set of the arm pose under the constraint of non-intersecting with the product models.

Different intersection detection methods are used for different types of products. Therefore, the product detection sequence should be determined by the efficiency of the corresponding intersection detection method. According to the efficiency from high to low, each product detected the intersection with the arm one by one. The intersection detection method efficiency will be studied in the next section.

All forearms in the initial solution set of the arm pose would detect the intersection with the product in the sequence. The forearms that intersect with the product would be

taken out at each step of the process. Similarly, the upper arms corresponding with the forearms that do not intersect with any product would be filtered. After the process, all arm poses that do not intersect with any product in the scene would constitute the reachable solution set of the arm pose.

The dimensionless value that the ratio of the quantity of the reachable solution set of the arm pose to the quantity of the initial solution set of the arm pose can derive can be called the degree of reachability (DOR). Its connotation is how much of all gestures the arm can make without obstacles, based on the hand being fixed, can reach the designated maintenance spot. The DOR can be used for the quantitative evaluation of the reachability of the maintenance spot.

### 3.3.3. Intersection Detection Efficiency Test

Considering the difference in computer performance, studying the intersection detection efficiency is necessary. In this study, efficiency is expressed in computational time. The size of the initial solution set of the arm pose and the intersection detection method are the main factors affecting efficiency.

By reasonably adjusting the division value n in the discretization process of the arm's maximum solution set, the different initial solution set of the arm pose that includes different quantity solutions can be obtained. Six objects, including one sphere, one cylinder, one capsule, two OBBs, and one triangular patch, are created for the efficiency test. Eight initial solution sets of the arm pose are obtained under the division values from 3° to 10°. Each initial solution set of the arm pose detected the intersection with these six objects three times (Table A1 in the Appendix A), meanwhile, the run time and DOR should also be recorded (Tables A2 and A3). Furthermore, by processing the data, the statistical parameters of the DOR under different division values can be obtained (Table A4). Based on the original test data above, the test results can be obtained in Figure 10.

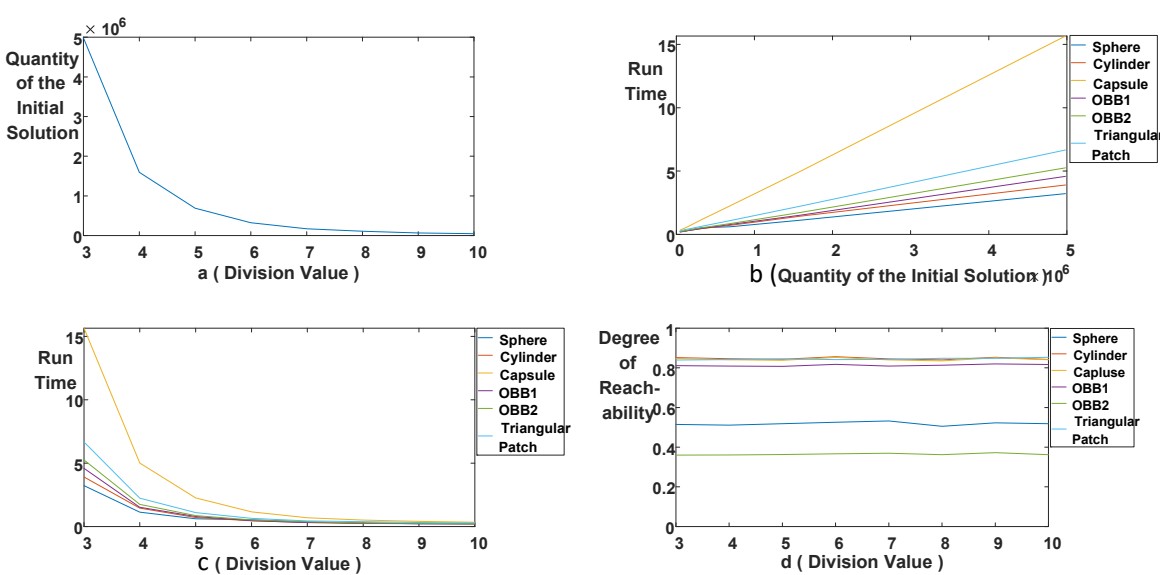

**Figure 10.** Results of the Test.

According to the figures above, the following analyses can be obtained.

(1)　As the division value increases, the quantity of initial solutions rapidly decreases.
(2)　The run time of the detection between line segments and different model objects and the quantity of initial solutions is linear.
(3)　Therefore, as the division value increases, the run time decreases rapidly.
(4)　The variation coefficients of the DOR obtained by different objects under different division values are below 2.5%. Little change was exhibited, as a larger division value can be chosen to solve the DOR.

(5) With the same initial solution set of arm poses, the run time of different intersection detection methods may also differ. The intersection detection efficiency ranges from high to low as line segment and sphere, cylinder, OBB, triangular patch, and capsule. In this test, efficiency is tested by one triangular patch, but the irregular model needs to be represented by several triangular patches. Therefore, according to this result, the detection order of different types of products is as follows: spheres, cylinders, OBBs, capsules, and irregular objects represented by triangular patches.

## 4. Case Study

### 4.1. Scene Construction

According to the methodology above, four scenes are constructed for analysis.

Three objects were used to test Scene 1, as shown in Figure 11a, which contains one sphere, one cylinder, and one OBB. Table 3 shows the necessary geometric characteristic data of these objects. In the table, for the sphere, the position parameter is its center coordinate and the range parameter is its radius. For the cylinder, the position parameter is its axis ends' coordinates and the range parameter is its radius. For OBB, the position parameter is its center coordinate, whereas the range parameter is a vector whose element is half the length of OBB's three sides and the direction parameter is direction vector of the three sides.

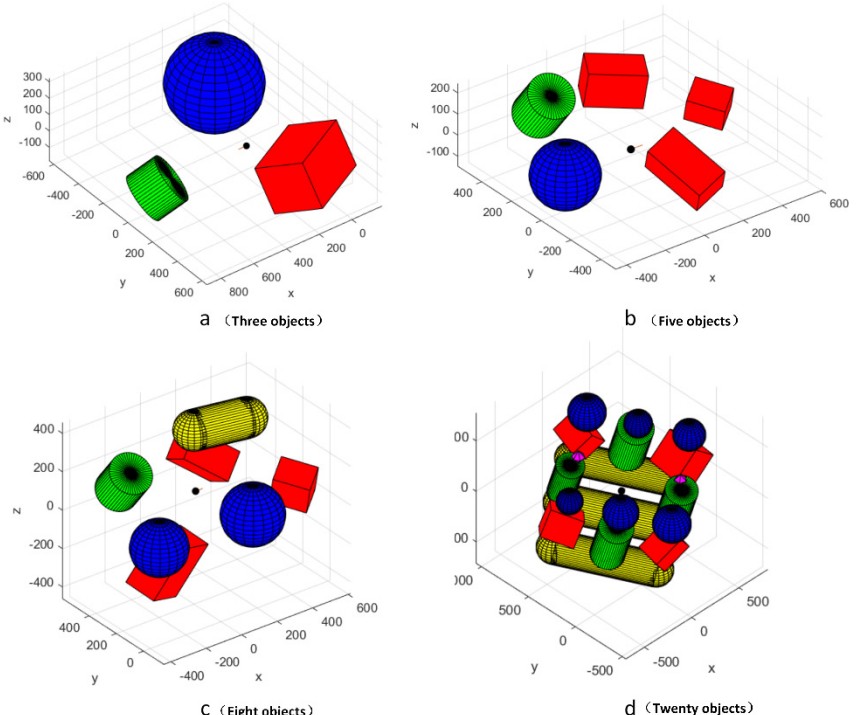

a （Three objects）  b （Five objects）

c （Eight objects）  d （Twenty objects）

**Figure 11.** Scene 1–Scene 4.

**Table 3.** Necessary Geometric Characteristic Data of the Object in Scene 1.

| Objects to Be Tested | Necessary Geometric Characteristic Data (mm) | | | Hand Coordinates (mm) |
| --- | --- | --- | --- | --- |
| | **Position Class** | **Range Class** | **Direction Class** | |
| Object 1 (Sphere) | (68.17, −387.15, 72.26) | 250.4977 | — | |
| Object 2 (Cylinder) | (796.61, 0.53, −73.66) (672.16, 64.31, 29.61) | 134.2806 | — | x = 217.24 y = 71.10 z = 27.97 |
| Object 3 (OBB) | (94.898, 388.26, 33.30) | [133.41, 140.89, 196.39] | u [-0.1902, 0.3570, −0.9145] v [0.4300, −0.8071, −0.4045] v [−0.8826, −0.4702, 0] | |

Five objects were used to test Scene 2, as shown in Figure 11b, which contains one sphere, one cylinder, and three OBBs. Table 4 shows the necessary geometric characteristic data of these objects, and the meanings are similar to those in Scene 1.

**Table 4.** Necessary Geometric Characteristic Data of the Object in Scene 2.

| Objects to Be Tested | Necessary Geometric Characteristic Data (mm) | | | Hand Coordinates (mm) |
|---|---|---|---|---|
| | Position Class | Range Class | Direction Class | |
| Object 1 (Sphere) | (−306.84, 33.05, 0) | 148.65 | — | |
| Object 2 (Cylinder) | (−128.13, 463.47, 23.52) (−145.23, 335.65, 176.75) | 102.64 | — | |
| Object 3 (OBB) | (240.31, 423.67, 73.945) | [124.49, 82.90, 73.95] | u [0.8708, −0.4763, −0.1214] v [−0.0529, 0.1547, −0.9865] w [−0.4887, −0.8655, −0.1095] | x = −14.76 y = −25.77 z = 45.49 |
| Object 4 (OBB) | (493.56, 115.57, 18.635) | [67.06, 63.16, 82.47] | u [0.1930, 0.3982, −0.8968] v [−0.7581, −0.5197, −0.3939] w [0.6229, −0.7559, −0.2015] | |
| Object 5 (OBB) | (28.5, −331.99, 77.8) | [70.58, 143.56, 77.80] | u [0.2798, 0.4915, −0.8247] v [0.2627, −0.8654, −0.4266] w [−0.9234, −0.0973, −0.3714] | |

Seven objects were used to test Scene 3, as shown in Figure 11c, which contains two spheres, one cylinder, one capsule, and three OBBs. Table 5 shows the necessary geometric characteristic data of these objects. For the capsule, the position parameter is its axis ends' coordinates and range parameter is its radius. For other objects, the parameter meanings are similar to those in Scene 1.

**Table 5.** Necessary Geometric Characteristic Data of the Object in Scene 3.

| Objects to Be Tested | Necessary Geometric Characteristic Data (mm) | | | Hand Coordinate (mm) |
|---|---|---|---|---|
| | Position Class | Range Class | Direction Class | |
| Object 1 (Sphere) | (−304.78, 58.23, 2.84) | 135.48 | — | |
| Object 2 (Sphere) | (174.60, −5.64, 37.27) | 148.65 | — | |
| Object 3 (Cylinder) | (−215.70, 383.47, 208.77) (−198.60, 511.29, 55.55) | 102.64 | — | |
| Object 3 (Capsule) | (85.13, 312.14, 361.81) (343.23, 258.90, 361.81) | 95.76 | — | |
| Object 5 (OBB) | (248.80, 457.52, 107.06) | [124.50, 82.90, 73.95] | u [0.7437, −0.3476, −0.5710] v [0.0847, 0.8964, −0.4352] w [-0.6631, −0.2752, −0.6961] | x = 74.02 y = 284.65 z = 60.80 |
| Object 6 (OBB) | (−103.40, 263.99, −242.31) | [142.10, 170.63, 77.80] | u [0.5516, −0.1250, 0.8247] v [0.7518, 0.5029, −0.4266] w [0.3613, −0.8553, −0.3713] | |
| Object 7 (OBB) | (490.63, 103.71, 30.84) | [87.15, 66.75, 90.91] | u [−0.6229, 0.7559, 0.2016] v [0.7581, 0.5196, 0.3939] w [0.1930, 0.3982, −0.8968] | |

Nineteen objects were tested in Scene 4, as shown in Figure 11d, which contains six spheres, four cylinders, three capsules, four OBBs, one irregular object represented by 10 triangular patches, and one irregular object represented by 12 triangular patches. Table 6 shows the necessary geometric characteristic data of these objects. For irregular objects represented by triangular patches, position parameter is the coordinates for the vertices and the direction parameter is the vertexes connection order. For other objects, the parameter meanings are similar to those in Scenes 1 and 3.

**Table 6.** Necessary Geometric Characteristic Data of the Object in Scene 4.

| Objects to Be Tested | Necessary Geometric Characteristic Data (mm) | | | | Hand Coordinates (mm) |
|---|---|---|---|---|---|
| | Position Class | | Range Class | Direction Class | |
| Object 1 (Sphere) | (22.21, 599.09, 630.79) | | 147.34 | — | |
| Object 2 (Sphere) | (515.04, 1.04, 471.33) | | 126.01 | — | |
| Object 3 (Sphere) | (−65.00, −396.52, 169.93) | | 147.69 | — | |
| Object 4 (Sphere) | (−557.83, 201.53, 329.40) | | 103.68 | — | |
| Object 5 (Sphere) | (268.62, 300.07, 551.06) | | 110.35 | — | |
| Object 6 (Sphere) | (−311.41, −97.50, 249.67) | | 138.75 | — | |
| Object 7 (Cylinder) | (427.98, 581.87, −13.35) | (337.05, 394.31, 409.04) | 140.84 | — | |
| Object 8 (Cylinder) | (372.07, 34.36, −268.50) | (281.14, −153.20, 153.89) | 126.45 | — | |
| Object 9 (Cylinder) | (−220.15, 137.65, −350.12) | (−311.08, −49.91, 72.27) | 134.72 | — | x = 58.45 |
| Object 10 (Cylinder) | (−164.24, 685.16, −94.97) | (−255.17, 497.60, 327.42) | 110.62 | — | y = 265.98 |
| Object 11 (Capsule) | (181.90, 928.48, −111.01) | (674.73, 330.43, −270.48) | 127.16 | — | z = 29.46 |
| Object 12 (Capsule) | (−398.14, 530.92, −412.41) | (94.69, −67.13, −571.87) | 135.13 | — | |
| Object 13 (Capsule) | (−108.12, 729.70, −261.71) | (384.71, 131.65, −421.18) | 147.82 | — | |
| Object 14 (OBB) | (102.05, 763.79, 259.89) | | [131.33, 104.72, 109.00] | u [0.1656, 0.9473, 0.2744]<br>v [0.6893, −0.3101, 0.6547]<br>w [0.7053, 0.0807, −0.7043] | |
| Object 15 (OBB) | (594.89, 165.74, 100.43) | | [126.94, 143.88, 146.31] | u [0.7549, 0.0344, −0.6549]<br>v [0.6547, −0.0990, 0.7494]<br>w [0.0391, 0.9945, 0.0972] | |
| Object 16 (OBB) | (14.85, −231.83, −200.97) | | [132.53, 100.72, 103.41] | u [0.8001, −0.1798, 0.5724]<br>v [0.5981, 0.1639, −0.7845]<br>w [0.0472, 0.9700, 0.2386] | |
| Object 17 (OBB) | (−477.99, 366.22, −41.51) | | [136.33, 114.72, 129.05] | u [0.5842, −0.7341, −0.3462]<br>v [0.5084, −0.0016, 0.8611]<br>w [0.6327, 0.6791, −0.3723] | |
| Object 18 (Irregular Object Represented by Triangular Patches) | P1 (225.02, −197.74, 370.88)<br>P2 (276.28, −197.74, 320.63)<br>P3 (240.86, −148.99, 320.63)<br>P4 (183.55, −167.61, 320.63) | P5 (183.55, −227.87, 320.63)<br>P6 (240.86, −246.49, 320.63)<br>P7 (225.02, −197.74, 270.38) | — | (P1, P2, P3) (P1, P6, P2) (P4, P5, P7)<br>(P1, P3, P4) (P2, P3, P7) (P5, P6, P7)<br>(P1, P4, P5) (P3, P4, P7) (P6, P2, P7)<br>(P1, P5, P6) | |
| Object 19 (Irregular Object Represented by Triangular Patches) | P1 (−267.81, 400.31, 541.67)<br>P2 (−214.75, 400.31, 480.10)<br>P3 (−241.28, 446.26, 480.10)<br>P4 (−294.34, 446.26, 480.10) | P5 (−320.87, 400.31, 480.10)<br>P6 (−294.34, 354.36, 480.10)<br>P7 (−241.28, 354.36, 480.10)<br>P8 (−267.81, 400.31, 418.53) | — | (P1, P2, P3) (P1, P6, P7) (P4, P5, P8)<br>(P1, P3, P4) (P1, P7, P2) (P5, P6, P8)<br>(P1, P4, P5) (P2, P3, P8) (P6, P7, P8)<br>(P1, P5, P6) (P3, P4, P8) (P7, P2, P8) | |

### 4.2. Calculation of DOR

In this section, the possible LRUs of the system are selected by the quantitative scoring to the minimum maintenance unit failure rate and complexity factors.

Failure Rate Factor

On the basis of the foregoing intersection detection efficiency test result, the DOR of Scenes 1 to 4 was calculated by using the initial solution set of arm poses obtained when the division value is 10°. The efficiency under the division value of 3° to 9° is also tested. Tables 7 and 8 show the run time and DOR of Scenes 1 to 4.

**Table 7.** Run Time (s) of Each Scene.

| Division Value | Quantity of the Initial Solution | Scene 1 | | Scene 2 | | Scene 3 | | Scene 4 | |
|---|---|---|---|---|---|---|---|---|---|
| | | Original Value | Average Value | Original Value | Average Value | Original Value | Average Value | Original Value | Average Value |
| 3° | 4,981,824 | 7.563694 | 7.540827 | 11.908870 | 11.86982 | 13.781394 | 14.091758 | 49.09117 | 48.913218 |
| | | 7.521761 | | 11.652367 | | 13.962062 | | 48.86793 | |
| | | 7.537027 | | 12.048212 | | 14.531819 | | 48.78056 | |
| 4° | 1,595,556 | 2.534249 | 2.511348 | 3.927523 | 3.884259 | 4.821261 | 4.743497 | 19.41575 | 19.420245 |
| | | 2.488741 | | 3.850726 | | 4.773427 | | 19.24142 | |
| | | 2.511055 | | 3.874527 | | 4.635803 | | 19.60356 | |
| 5° | 690,954 | 1.215654 | 1.198221 | 1.891283 | 1.857310 | 2.462933 | 2.432991 | 11.46066 | 11.461423 |
| | | 1.190019 | | 1.878364 | | 2.436729 | | 11.47515 | |
| | | 1.188991 | | 1.802282 | | 2.399312 | | 11.44846 | |
| 6° | 321,408 | 0.736962 | 0.689658 | 1.065829 | 1.015121 | 1.410491 | 1.374507 | 8.246137 | 8.243521 |
| | | 0.679460 | | 0.973468 | | 1.351717 | | 8.265693 | |
| | | 0.652551 | | 1.006067 | | 1.361314 | | 8.218733 | |
| 7° | 169,533 | 0.514386 | 0.492571 | 0.700348 | 0.683232 | 0.869385 | 0.883408 | 6.866653 | 6.916059 |
| | | 0.477544 | | 0.663157 | | 0.860397 | | 6.985432 | |
| | | 0.485783 | | 0.686192 | | 0.920443 | | 6.896092 | |
| 8° | 108,864 | 0.389968 | 0.378775 | 0.581512 | 0.576356 | 0.742773 | 0.724335 | 6.280376 | 6.295347 |
| | | 0.364600 | | 0.610199 | | 0.70081 | | 6.320756 | |
| | | 0.381758 | | 0.537356 | | 0.729423 | | 6.284908 | |
| 9° | 66,528 | 0.315801 | 0.297481 | 0.449422 | 0.432285 | 0.556314 | 0.558371 | 5.995492 | 5.282333 |
| | | 0.294360 | | 0.429213 | | 0.573013 | | 6.030587 | |
| | | 0.282283 | | 0.418219 | | 0.545785 | | 5.971768 | |
| 10° | 48,450 | 0.291584 | 0.287173 | 0.404568 | 0.408200 | 0.511533 | 0.527139 | 5.651172 | 5.670438 |
| | | 0.265109 | | 0.391777 | | 0.528807 | | 5.760809 | |
| | | 0.304827 | | 0.428255 | | 0.541078 | | 5.599334 | |

**Table 8.** DOR (dimensionless) of Each Scene.

| Division Value (Quantity of the Initial Solution) | Scene 1 | Scene 2 | Scene 3 | Scene 4 |
|---|---|---|---|---|
| 3° (4,981,824) | 0.660797 | 0.670268 | 0.176648 | 0.148202 |
| 4° (1,595,556) | 0.663554 | 0.672205 | 0.178140 | 0.151271 |
| 5° (690,954) | 0.662927 | 0.671952 | 0.178584 | 0.152019 |
| 6° (321,408) | 0.661381 | 0.673322 | 0.177351 | 0.147205 |
| 7° (169,533) | 0.664514 | 0.666684 | 0.177659 | 0.157727 |
| 8° (108,864) | 0.665812 | 0.676495 | 0.181915 | 0.152732 |
| 9° (66,528) | 0.658475 | 0.673626 | 0.180435 | 0.148855 |
| 10° (48,450) | 0.67356 | 0.676533 | 0.176718 | 0.152219 |

By processing the data in Table 8, the statistical parameters of the DOR under different division values can be obtained.

According to the data in Table 9, the following analyses can be obtained.

**Table 9.** Statistical Parameters of the DOR of Each Scene Obtained under Different Division Values.

| Scene | Average Value | Variance | Standard Deviation | Variable Coefficient |
|---|---|---|---|---|
| Scene 1 | 0.66387750 | 0.00001798 | 0.00423992 | 0.00638660 |
| Scene 2 | 0.67263563 | 0.00000915 | 0.00302524 | 0.00449759 |
| Scene 3 | 0.17843125 | 0.00000302 | 0.00173703 | 0.00973501 |
| Scene 4 | 0.15127875 | 0.00000963 | 0.00310363 | 0.02051600 |

1. The irregular objects represented by triangular patches have a significant influence on run time.
2. The variation coefficients of DOR obtained by different scenes under different division values are below 2.5%. The result further proves that the division values have little influence on the DOR, as a relatively large division value can be chosen to solve the DOR.

**5. Conclusions and Discussion**

In this study, an automatic and quantitative approach is proposed on the basis of human arm spatial data to conduct maintenance reachability. First, an arm model that considers kinematics is established to obtain the formulae of the joint positions in the hand coordinate system, then the initial solution set of the arm pose was generated by dividing the joint space. Second, to enhance the calculation efficiency, a 3D model reconstruction method is proposed. The reconstructed models are further parameterized for convenient calculation. Moreover, several intersection detection methods are adopted purposefully for various objects with different geometrical characteristics in the maintenance scene. Third, the impact of the two points above, namely, different joint range divisions and the intersection detection method selection on the efficiency, are analyzed. Fourth, in the analysis, the DOR is introduced to quantify the extent of reachability, and the impact of different joint range divisions on DOR are also analyzed. The conclusion of the analysis can be summarized as the intersection detection method efficiency ranges from high to low as the line segment and sphere, cylinder, OBB, triangular patch, and capsule. As the division value decreases, the efficiency increases rapidly and the division value has little effect on DOR. Lastly, four cases are selected to verify the methodology.

The advantages of the proposed methodology can be expressed as follows.

(1) All possible arm postures that are required to move freely are considered and transformed into an initial global dataset, whereas the subsequent analysis is conducted by starting from this initial global dataset. Consequently, the finitude occurring in the current methods can be reduced to some extent.
(2) The determination of the initial global data set is also fully optimized and screened, which not only ensures the data of the arm are not lost, but also ensures the efficiency of the calculation process.
(3) As the calculation process and resulting expression of reachability are quantitative, the judgments on whether the maintenance spot is reachable or not and how easy it is to reach have a lower dependence on a person. Hence, the proposed methodology is objective.
(4) On the basis of quantitative analysis, the proposed can not only analyze the quality of accessibility design, but also show how good or how bad it is by means of quantitative expression.

However, the proposed methodology requires improvements. Owing to the importance of the initial hand posture in the method, considerations on hand posture in maintenance should be covered. Future works will focus on this point to enhance the adaptability

in the complex maintenance scene. Moreover, the proposed can be also integrated into a product design platform to improve its usability.

**Author Contributions:** Conceptualization, C.L.; Methodology, J.G.; Validation, Y.L.; Investigation, H.G.; Writing—original draft, H.Z. All authors have read and agreed to the published version of the manuscript.

**Funding:** The work is supported by the National Natural Science Foundation of China (71701005).

**Conflicts of Interest:** The authors declare no conflict of interest.

## Appendix A

**Table A1.** Necessary Geometric Characteristic Data of Objects to Test the Hand Coordinates.

| Objects to Be Tested | Necessary Geometric Characteristic Data (mm) | | | Hand Coordinates (mm) |
|---|---|---|---|---|
| | **Position Class** | **Range Class** | **Direction Class** | |
| Object 1 (Sphere) | (100, 100, 200) | 244.9489 | — | x = 100 y = 100 z = 475 |
| Object 2 (Cylinder) | (40.7, 45.2, 63.4) (182.2, 156.6, 187.3) | 75.7 | — | x = 225.1 y = 183.7 z = 216.2 |
| Object 3 (Capsule) | (40.7, 45.2, 63.4) (182.2, 156.6, 187.3) | 75.7 | — | x = 275.1 y = 233.7 z = 256.2 |
| Object 4 (OBB1) | (91, 28, 75) | [145, 292, 274] | u [0.4773, 0.6765, −0.5608] v [0.8208, −0.5712, 0.0096] w [0.3139, 0.4649, 0.8279] | x = 346.4 y = 317.7 z = 335.6 |
| Object 5 (OBB2) | (100, 100, 200) | [244.95, 244.95, 244.95] | [1, 0, 0] [0, 1, 0] [0, 0, 1] | x = 100 y = 100 z = 475 |
| Object 6 (Triangular Patch) | P1 (456.48, 433.09, 355.99) P2 (440.12, 390.40, −31.32) P3 (292.01, 15.38, 16.27) | — | (P1, P2, P3) | x = 81 y = 90 z = 12 |

**Table A2.** Run Time (s) Comparison of Intersection Detection Under Different Division Values.

| Division Value | Quantity of the Initial Solution | Object 1 Sphere | | Object 2 Cylinder | | Object 3 Capsule | | Object 4 OBB1 | | Object 5 OBB2 | | Object 6 Triangular Patch | |
|---|---|---|---|---|---|---|---|---|---|---|---|---|---|
| | | Original Value | Average Value | Original Value | Average Value | Original Value | Average Value | Original Value | Average Value | Original Value | Average Value | Original Value | Average Value |
| 3° | 4,981,824 | 3.2870 | | 3.9342 | | 15.8284 | | 4.598788 | | 5.2575 | | 6.6144 | |
| | | 3.2535 | 3.2203 | 3.8897 | 3.905560 | 15.4594 | 15.654 | 4.529736 | 4.5872 | 5.2467 | 5.2483 | 6.6569 | 6.6653 |
| | | 3.1202 | | 3.8926 | | 15.6754 | | 4.633236 | | 5.2406 | | 6.7244 | |
| 4° | 1,595,556 | 1.1313 | | 1.3852 | | 4.9639 | | 1.513430 | | 1.7770 | | 2.2744 | |
| | | 1.0894 | 1.1272 | 1.5448 | 1.448390 | 4.9284 | 5.0094 | 1.532902 | 1.5208 | 1.7374 | 1.7440 | 2.2341 | 2.2388 |
| | | 1.1610 | | 1.4150 | | 5.1361 | | 1.516288 | | 1.7176 | | 2.2079 | |
| 5° | 690,954 | 0.6418 | | 0.7491 | | 2.2695 | | 0.787951 | | 0.8753 | | 1.1138 | |
| | | 0.5722 | 0.6041 | 0.6791 | 0.713749 | 2.2601 | 2.2613 | 0.860389 | 0.7901 | 0.8434 | 0.8543 | 1.0693 | 1.0937 |
| | | 0.5982 | | 0.7129 | | 2.2542 | | 0.722244 | | 0.8441 | | 1.0979 | |
| 6° | 321,408 | 0.4115 | | 0.4676 | | 1.1390 | | 0.463332 | | 0.5616 | | 0.6303 | |
| | | 0.4155 | 0.5278 | 0.4521 | 0.452351 | 1.1631 | 1.1551 | 0.439418 | 0.4627 | 0.4862 | 0.5227 | 0.6468 | 0.6390 |
| | | 0.7565 | | 0.4372 | | 1.1632 | | 0.485351 | | 0.5203 | | 0.6399 | |
| 7° | 169,533 | 0.4939 | | 0.3207 | | 0.7050 | | 0.323831 | | 0.3390 | | 0.4552 | |
| | | 0.3080 | 0.3581 | 0.2939 | 0.313811 | 0.6865 | 0.6882 | 0.302256 | 0.3151 | 0.3322 | 0.3652 | 0.4367 | 0.4419 |
| | | 0.2724 | | 0.326733 | | 0.673141 | | 0.319245 | | 0.424404 | | 0.4337 | |
| 8° | 108,864 | 0.2366 | | 0.2852 | | 0.5276 | | 0.2859 | | 0.2848 | | 0.3686 | |
| | | 0.2424 | 0.2416 | 0.2450 | 0.2635 | 0.4789 | 0.5032 | 0.2598 | 0.2720 | 0.2756 | 0.2804 | 0.3537 | 0.3789 |
| | | 0.2457 | | 0.2604 | | 0.5032 | | 0.2702 | | 0.2808 | | 0.4145 | |
| 9° | 66,528 | 0.2237 | | 0.2100 | | 0.4062 | | 0.2384 | | 0.2773 | | 0.3335 | |
| | | 0.2109 | 0.2189 | 0.2145 | 0.2175 | 0.3934 | 0.4003 | 0.2151 | 0.2214 | 0.2389 | 0.2464 | 0.2929 | 0.3100 |
| | | 0.2221 | | 0.2279 | | 0.4015 | | 0.2108 | | 0.2231 | | 0.3037 | |
| 10° | 48,450 | 0.1981 | | 0.1888 | | 0.3139 | | 0.2141 | | 0.2224 | | 0.3202 | |
| | | 0.2241 | 0.2081 | 0.2063 | 0.1956 | 0.3046 | 0.3313 | 0.2304 | 0.2204 | 0.2089 | 0.2233 | 0.3121 | 0.2769 |
| | | 0.2020 | | 0.1917 | | 0.3754 | | 0.2168 | | 0.2387 | | 0.3604 | |

**Table A3.** DOR (dimensionless) Comparison Under Different Division Values.

| Division Value | Quantity of the Initial Solution | Objects to Be Tested | | | | | |
|---|---|---|---|---|---|---|---|
| | | Object 1 Sphere | Object 2 Cylinder | Object 3 Capsule | Object 4 OBB1 | Object 5 OBB2 | Object 6 Triangular Patch |
| 3° | 4,981,824 | 0.514630 | 0.851506 | 0.847068 | 0.810667 | 0.359932 | 0.839534 |
| 4° | 1,595,556 | 0.510923 | 0.845773 | 0.840718 | 0.809181 | 0.360957 | 0.841159 |
| 5° | 690,954 | 0.518665 | 0.842771 | 0.837923 | 0.807433 | 0.362903 | 0.845120 |
| 6° | 321,408 | 0.525805 | 0.855813 | 0.852431 | 0.817802 | 0.366680 | 0.841382 |
| 7° | 169,533 | 0.531826 | 0.845334 | 0.840238 | 0.808893 | 0.369285 | 0.842119 |
| 8° | 108,864 | 0.504896 | 0.840204 | 0.835198 | 0.812895 | 0.362030 | 0.848067 |
| 9° | 66,528 | 0.522953 | 0.852829 | 0.850078 | 0.819429 | 0.372490 | 0.848289 |
| 10° | 48,450 | 0.518514 | 0.841940 | 0.840475 | 0.816429 | 0.362270 | 0.853437 |

**Table A4.** Statistical Parameters of the DOR of Each Object Obtained under Different Division Values.

| Objects to Be Tested | Average Value | Variance | Standard Deviation | Variable Coefficient |
|---|---|---|---|---|
| Object 1 Sphere | 0.51852650 | 0.00006353 | 0.00797067 | 0.01537177 |
| Object 2 Cylinder | 0.84702125 | 0.00002824 | 0.00531391 | 0.00627365 |
| Object 3 Capsule | 0.84301613 | 0.00003268 | 0.00571672 | 0.00678127 |
| Object 4 OBB1 | 0.81284113 | 0.00001798 | 0.00424036 | 0.00521671 |
| Object 5 OBB2 | 0.36456838 | 0.00001731 | 0.00416075 | 0.01141282 |
| Object 6 Triangular Patch | 0.84488838 | 0.00001967 | 0.00443484 | 0.00524902 |

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
