# Peer review of "Spatial Data-Based Automatic and Quantitative Approach in Analyzing Maintenance Reachability"

_applsci, doi:10.3390/app122412804_

Round 1
Reviewer 1 Report
The title reflects well the content of the article
The abstract is concise but unnecessarily long, I recommend shortening it.
The introduction introduces the reader to the issue and sets the article appropriately in the broader context.
In the literature review, I would also recommend devoting a paragraph to Maintenance in general and other disciplines in a broader context, and then only after that focus on virtual maintenance. Here I would recommend sources:
Achouch, M.; Dimitrova, M.; Ziane, K.; Sattarpanah Karganroudi, S.; Dhouib, R.; Ibrahim, H.; Adda, M. On Predictive Maintenance in Industry 4.0: Overview, Models, and Challenges. Appl. Sci. 2022, 12, 8081. doi: 10.3390/app12168081
Pech, M.; Vrchota, J.; Bednar, J. Predictive Maintenance and Intelligent Sensors in Smart Factory: a review. Sensors 2021, 21, 1470, doi:10.3390/s21041470.
Holstein, S.A.; Suman, V.J.; Hillengass, J.; McCarthy, P.L. Future Directions in Maintenance Therapy in Multiple Myeloma. J. Clin. Med. 2021, 10, 2261. doi: 10.3390/jcm10112261
Line 452 ERROR
The Methodology section is exciting and well described, but there is often an overlap between the Methodology and Practice sections of the authors' research. I recommend that the Methodology chapter be more clearly defined. And create a new chapter where all the data and conclusions of the authors will be presented. The case study chapter is well conceived.
Table 1 contains unnecessary information that the reader does not need to work with, I recommend that it be condensed. Just as it is unnecessary to burden the reader with calculations and matrix editing. The main contributions of the paper are elsewhere.
In addition to the results and the case study, I positively evaluate the conclusions, including mentioning the advantages and outlining the limitations and future direction of the paper.
Reviewer 2 Report
The paper is interesting, but can be improved. The abstrac is too long and can be improved.
Reviewer 3 Report
The paper proposes an interesting methodology in order to evaluate reachability based on an automatic and quantitative approach based on the spatial data of the human arm. The authors intend to contribute to the literature showing the approach and its application in practice, improving the maintenance reachability analysis. Although the topic is quite specific, maintenance tasks analysis is a very relevant topic that has received quite a lot of attention recently both in academia and in practice. Although exists a lot of different papers in the area, this paper uses an innovative appraoch that reduces the analysis subjectivity. The research questions is clear, and the research method selected is appropriate for the type of research conducted.
The research background is relevant but could be enlarged citing other papers that focuses on reachability. I suggest to cite other relevant papers expecially in the introduction part, in which there are many assumpions not supported by citation. Moreover, I suggest to include Digital Human Modeling sw as a tool to conduct reachability analysis (for example, see An automatic procedure based on virtual ergonomic analysis to promote human-centric manufacturing). The research workflow is well described and there is a rich paragraph that describes in deep the work supported by formulas. Results are presented in a clear way and are easily interpretable ,but could be deeper discussed. Good language and good academic way of writing. In my opinion, the abstract could be shortened. I suggest to review Fig.1 because is not very clear, maybe with less colours the figure could be more comprehensible. Overall, the paper could be accepted with minor revisions.
